# Ligand recognition and gating mechanism through three ligand-binding sites of human TRPM2 channel

**Yihe Huang, Becca Roth, Wei Lü\*, Juan Du\***

Van Andel Institute, Grand Rapids, United States

**Abstract** TRPM2 is critically involved in diverse physiological processes including core temperature sensing, apoptosis, and immune response. TRPM2's activation by $Ca^{2+}$ and ADP ribose (ADPR), an $NAD^+$-metabolite produced under oxidative stress and neurodegenerative conditions, suggests a role in neurological disorders. We provide a central concept between triple-site ligand binding and the channel gating of human TRPM2. We show consecutive structural rearrangements and channel activation of TRPM2 induced by binding of ADPR in two indispensable locations, and the binding of $Ca^{2+}$ in the transmembrane domain. The 8-Br-cADPR—an antagonist of cADPR—binds only to the MHR1/2 domain and inhibits TRPM2 by stabilizing the channel in an apo-like conformation. We conclude that MHR1/2 acts as a orthostatic ligand-binding site for TRPM2. The NUDT9-H domain binds to a second ADPR to assist channel activation in vertebrates, but not necessary in invertebrates. Our work provides insights into the gating mechanism of human TRPM2 and its pharmacology.
DOI: https://doi.org/10.7554/eLife.50175.001

## Introduction

ADP ribose (ADPR) and its derivatives, which are products of $NAD^+$ metabolism, play essential roles in neurotoxicity and cellular signaling under many physiological and pathological conditions (*Ernst et al., 2013*; *Gasser et al., 2006*; *Guse, 2015*; *Kolisek et al., 2005*; *Nikiforov et al., 2015*). In neurodegenerative diseases, the progressive accumulation of poly-ADPR and free ADPR results from poly(adenosine 5'-diphosphate-ribose) polymerase-1 activation, and contributes to neurotoxicity and neuronal death (*Kam et al., 2018*). Intracellular ADPR targets and activates the nonselective, calcium-permeable TRPM2 channel in the presence of $Ca^{2+}$. ADPR acts on one hand as a secondary messenger that contributes to insulin secretion, redox, and temperature sensation, but on the other hand, its accumulation results in intracellular calcium overload and eventually apoptosis (*Hara et al., 2002*; *Hecquet et al., 2014*; *Song et al., 2016*; *Tan and McNaughton, 2016*; *Togashi et al., 2006*; *Uchida et al., 2011*; *Wehage et al., 2002*). TRPM2 is thus considered to be a key contributor in brain injury via a glutamate-independent pathway in neurodegenerative conditions or after stroke, and it has important functions in obesity, diabetes, bipolar disorder, and Alzheimer's disease (*Aminzadeh et al., 2018*; *Jang et al., 2015*; *Sita et al., 2018*; *Uchida and Tominaga, 2014*; *Zhang et al., 2012*).

Belonging to the transient receptor potential melastatin family (TRPM), TRPM2 shares a characteristic MHR1-4 domain, a transmembrane domain (TMD), and a coiled-coil domain with the other family members. It is unique due to owning a characteristic C-terminal NUDT9-H domain, which has been considered a classic binding site for ADPR and its derivatives (*Fliegert et al., 2017*; *Perraud et al., 2001*), but the presence of the ADPR binding site has not been directly observed. Recently, our group defined a novel ADPR binding site in the MHR1/2 domain that is essential for the activation of the zebrafish TRPM2 (*dr*TRPM2) channel (*Huang et al., 2018*). It remains unclear

**\*For correspondence:**
wei.lu@vai.org (WLü);
juan.du@vai.org (JD)

**Competing interests:** The authors declare that no competing interests exist.

whether the MHR1/2 domain is a universal binding site across all species; whether the NUDT9-H domain has a second ADPR binding site; and what the functional role of the NUDT9-H domain is. Beyond the agonist ADPR, TRPM2 also responds to a variety of ADPR orthologues including the agonist cyclic ADPR (cADPR) (*Kolisek et al., 2005*; *Yu et al., 2019*), which is produced during acute kidney injury (*Kolisek et al., 2005*; *Togashi et al., 2006*), and the antagonist 8-bromo-cyclic ADP-ribose (8-Br-cADPR), which decreases ischemia-reperfusion injury when used in treatment (*Eraslan et al., 2019*). Interestingly, 8-Br-cADPR is an antagonist for cADPR but does not inhibit ADPR-evoked currents (*Kolisek et al., 2005*). Despite extensive studies (*Baszczyňski et al., 2019*; *Eraslan et al., 2019*; *Fliegert et al., 2018*; *Fourgeaud et al., 2019*; *Kolisek et al., 2005*; *Kühn et al., 2019*; *Moreau et al., 2013*; *Tóth and Csanády, 2010*), there is limited knowledge about the TRPM2 drug binding sites and how drugs manipulate channel function.

Human TRPM2 (*hs*TRPM2) has several functional properties different from those of *dr*TRPM2 and invertebrate TRPM2, including temperature sensitivity and desensitization rate (*Nam Tran et al., 2018*; *Iordanov et al., 2019*). A recent publication suggested that *hs*TRPM2 does not bind ADPR in the MHR1/2 domain and that its activation is rather driven by ADPR binding in the NUDT9-H domain (*Wang et al., 2018*). However, this suggestion was based on a cryo-electron microscopy (cryo-EM) structure of *hs*TRPM2 in the presence of ADPR and $Ca^{2+}$ at 6.4 Å, a resolution that cannot clarify whether ADPR is present in either the MHR1/2 or the NUDT9-H domain. Moreover, the assignment of the functional states of the structures in this paper is not clear because there is a lack of side-chain densities in the ADPR/$Ca^{2+}$-bound structure and because the transmembrane domain (TMD) in the apo structure is barely visible.

To understand the mechanism underlying agonist and antagonist recognition, and gating mechanism of human TRPM2, we used single-particle cryo-EM to determine four full-length *hs*TRPM2 structures: in the apo state in the presence of EDTA (EDTA-*hs*TRPM2); in a non-activated state in complex with the agonist ADPR (ADPR-*hs*TRPM2); in a pre-open or an inactive state in the presence of ADPR/$Ca^{2+}$ (ADPR/$Ca^{2+}$-*hs*TRPM2); and in an inhibited state with the cADPR antagonist 8-Br-cADPR in the presence of $Ca^{2+}$ (8-Br-cADPR/$Ca^{2+}$-*hs*TRPM2), at 3.3, 4.4, 3.7, and 3.7 Å, respectively (*Figure 1—figure supplements 1* and *2*).

## Results

### Structure determination of *hs*TRPM2

In our four *hs*TRPM2 structures, the densities for nearly the entire intracellular domain are well-defined, including the NUDT9-H domain and the β1-β2 loop of the MHR1 domain (*Figure 1a–d*, *Figure 1—figure supplement 3*), which were poorly defined or unresolved in the *dr*TRPM2 structures (*Huang et al., 2018*). In both ADPR-bound structures, ADPR densities were observed in two different sites on each subunit, one (ADPR1) in the cleft of the MHR1/2 domain and a second (ADPR2) in the cleft of the NUDT9-H domain (*Figure 1b,c,f,g*; *Figure 1—figure supplement 4*). In contrast, we found a ring-shaped 8-Br-cADPR density only in the cleft of the MHR1/2 domain of the 8-Br-cADRP/$Ca^{2+}$-*hs*TRPM2 structure (*Figure 1d,h*; *Figure 1—figure supplement 4*). In addition, $Ca^{2+}$ density was visible in the ADPR/$Ca^{2+}$- and 8-Br-cADPR/$Ca^{2+}$-*hs*TRPM2 maps (*Figure 1g,h*), in the same location as in the *hs*TRPM4, *Ficedula albicollis* TRPM8 (*fa*TRPM8), *dr*TRPM2, and *Nematostella vectensis* TRPM2 (*nv*TRPM2) structures (*Autzen et al., 2018*; *Huang et al., 2018*; *Yin et al., 2019a*; *Zhang et al., 2018*).

While the majority of the TMD is also well-defined in the four *hs*TRPM2 structures (*Figure 1—figure supplement 3e–h*), less well resolved are the extracellular half of the pore-lining helix S6, the pore helix/loop, and the extracellular loops, similar to those in the TRPM8 structures (*Yin et al., 2019a*; *Yin et al., 2018*). Nevertheless, we were able to trace the protein backbone in the S6 helix. We speculate that the poorly defined pore region is caused, at least partially, by β-mercaptoethanol, which is essential for stabilizing the protein during purification but may reduce the conserved disulfide bond formed by C996 and C1008 in the extracellular loops that is important for channel function (*Jang et al., 2019*; *Mei et al., 2006*; *Mittal et al., 2017*; *Mittal et al., 2015*). That bond is a key interaction that stabilizes the integrity of the pore region in the *dr*TRPM2 structure (*Huang et al., 2018*).

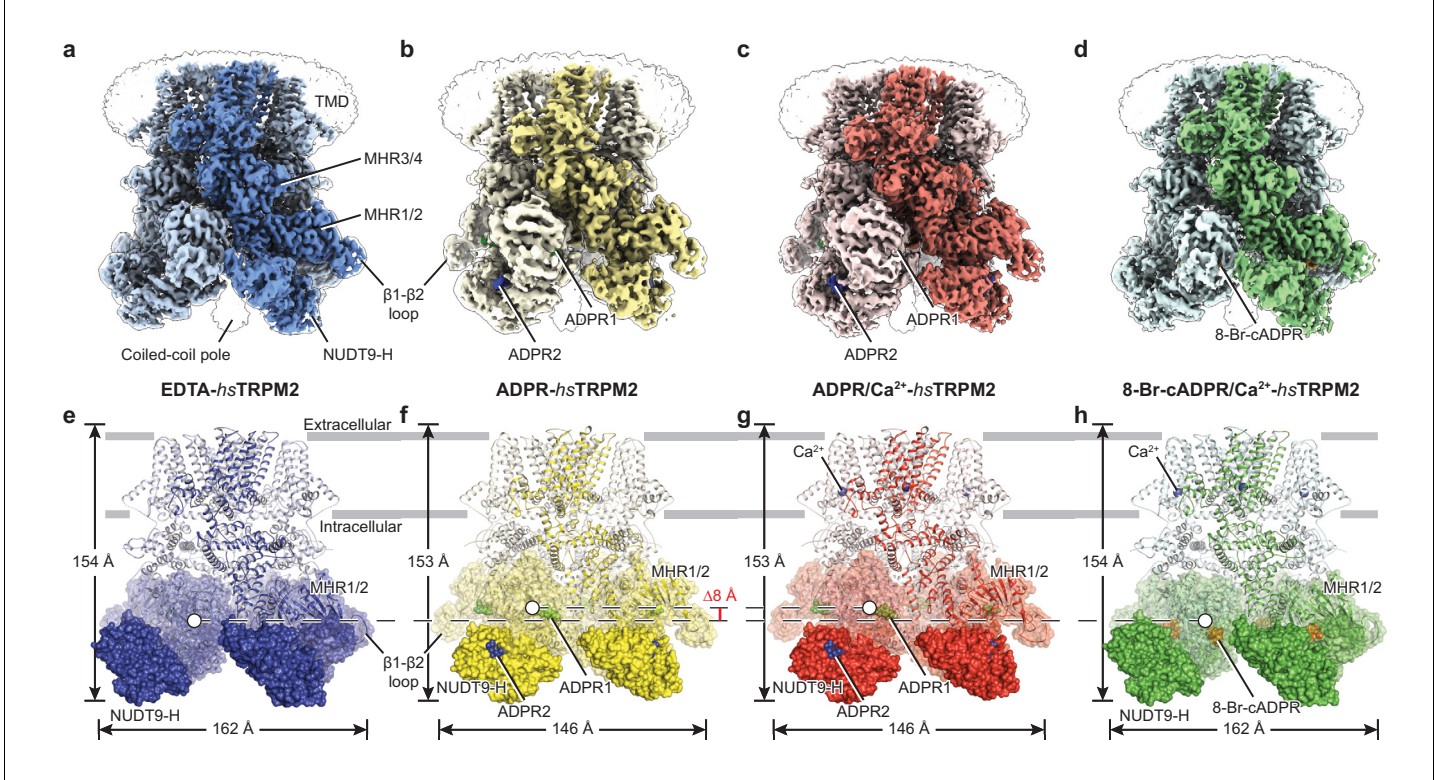

**Figure 1.** The overall architecture of *hs*TRPM2. The three-dimensional reconstructions of (**a**) EDTA-*hs*TRPM2, (**b**) ADPR-*hs*TRPM2, (**c**) ADPR/Ca$^{2+}$-*hs*TRPM2, and (**d**) 8-Br-cADPR/Ca$^{2+}$-*hs*TRPM2. The unsharpened reconstructions are shown as transparent envelopes. One subunit is highlighted. (**e–h**) Atomic models of the corresponding reconstructions in panels a-d. The NUDT9-H domains are shown in solid surface, the MHR1/2 domains are shown in transparent surface, and the rest of the proteins are shown in cartoon representation. Ca$^{2+}$, ADPR1, ADPR2, and 8-Br-cADPR are shown in purple, green, blue, and orange spheres, respectively. The center-of-mass (COM) of the MHR1/2 domain of one subunit in each structure is shown as a circle filled with white. The dimensions of the proteins and the difference between COMs along the pore axis are labeled.
DOI: https://doi.org/10.7554/eLife.50175.002

The following figure supplements are available for figure 1:

**Figure supplement 1.** The cryo-EM data processing flowchart for *hs*TRPM2 using the data of ADPR/Ca$^{2+}$-*hs*TRPM2 as an example.
DOI: https://doi.org/10.7554/eLife.50175.003

**Figure supplement 2.** Cryo-EM data analysis of *hs*TRPM2.
DOI: https://doi.org/10.7554/eLife.50175.004

**Figure supplement 3.** Local resolution estimation and representative densities of *hs*TRPM2 structures.
DOI: https://doi.org/10.7554/eLife.50175.005

**Figure supplement 4.** Ligand densities.
DOI: https://doi.org/10.7554/eLife.50175.006

**Figure supplement 5.** Comparison of the structures of *hs*TPPM2, *dr*TRPM2, and *nv*TRPM2.
DOI: https://doi.org/10.7554/eLife.50175.007

## Overall architecture and structural comparison

The *hs*TRPM2 structures share a four-layer arrangement with *dr*TRPM2 (*Huang et al., 2018*; *Yin et al., 2019b*), having a TMD, MHR3/4, and a ligand-sensing layer that includes the MHR1/2 and NUDT9-H domains, from top to bottom. However, their overall shapes differ, primarily because the NUDT9-H domain is positioned differently. In the EDTA-*hs*TRPM2 structure (*Figure 1e*, *Figure 1— figure supplement 5a*), the NUDT9-H domain is clamped between cognate and adjacent MHR1/2 domains, forming both intra- and intersubunit interfaces, while only the intrasubunit interface exists in the EDTA-*dr*TRPM2 structure (*Figure 1—figure supplement 5c*). As a consequence, the NUDT9-H domain forms a compact structure interacting with the adjacent MHR1/2 domain in *hs*TRPM2, whereas it hangs freely in the *dr*TRPM2 on the bottom of the protein. Moreover, the NUDT9-H domain is completely invisible in the *nv*TRPM2 structure (*Figure 1—figure supplement*

*5e*), despite high resolution of the rest of the protein (*Zhang et al., 2018*). Such an incrementally tighter coupling between the NUDT9-H domain and the rest of the protein, from invertebrate TRPM2 to human TRPM2, is in line with previous reports that the NUDT9-H domain plays an important role in channel gating of *hs*TRPM2 and *dr*TRPM2, but does not affect the channel gating of *nv*TRPM2 (*Fliegert et al., 2017*; *Kühn and Lückhoff, 2004*; *Wehage et al., 2002*; *Yu et al., 2017*).

Relative to the apo state (*Figure 1e*), the binding of ADPR to *hs*TRPM2 yielded a markedly elevated MHR1/2 layer and a contracted NUDT9-H layer, with the NUDT9-H domain swinging toward the pore axis (*Figure 1f–g*). ADPR molecules were identified, with ADPR1 in the MHR1/2 domain and ADPR2 in the NUDT9-H domain (*Figure 2a,c–f*). Despite a long-existing consensus view that the NUDT9-H domain binds ADPR, this is the first time that the ADPR density has been visualized in this location (*Figure 2d*). In addition, the clearly defined ADPR1 in the MHR1/2 domain is consistent with the one observed in the *dr*TRPM2 structure (*Figure 1—figure supplement 5b,d*) (*Huang et al., 2018*). Given that key residues in the MHR1/2 site are conserved across species (*Figure 2—figure supplement 1*) and that the invertebrate *nv*TRPM2 responds to ADPR independent of their NUDT9-H domains (*Kühn et al., 2016*), our data strongly supports the concept that that the ligand-binding site in MHR1/2 is conserved throughout all species.

In contrast to ADPR binding, the binding of 8-Br-cADPR did not produce obvious changes in the overall shape of the protein relative to the apo state, in agreement with the view that a competitive antagonist inhibits the protein by stabilizing it in an apo-like state (*Figure 1a,d,e,h*; *Figure 2b,g*). Similar mechanism has been observed for human P2 × 3 receptor, in which the competitive antagonists TNP-ATP and A-317491 stabilize the apo/resting state (*Mansoor et al., 2016*). Surprisingly, in

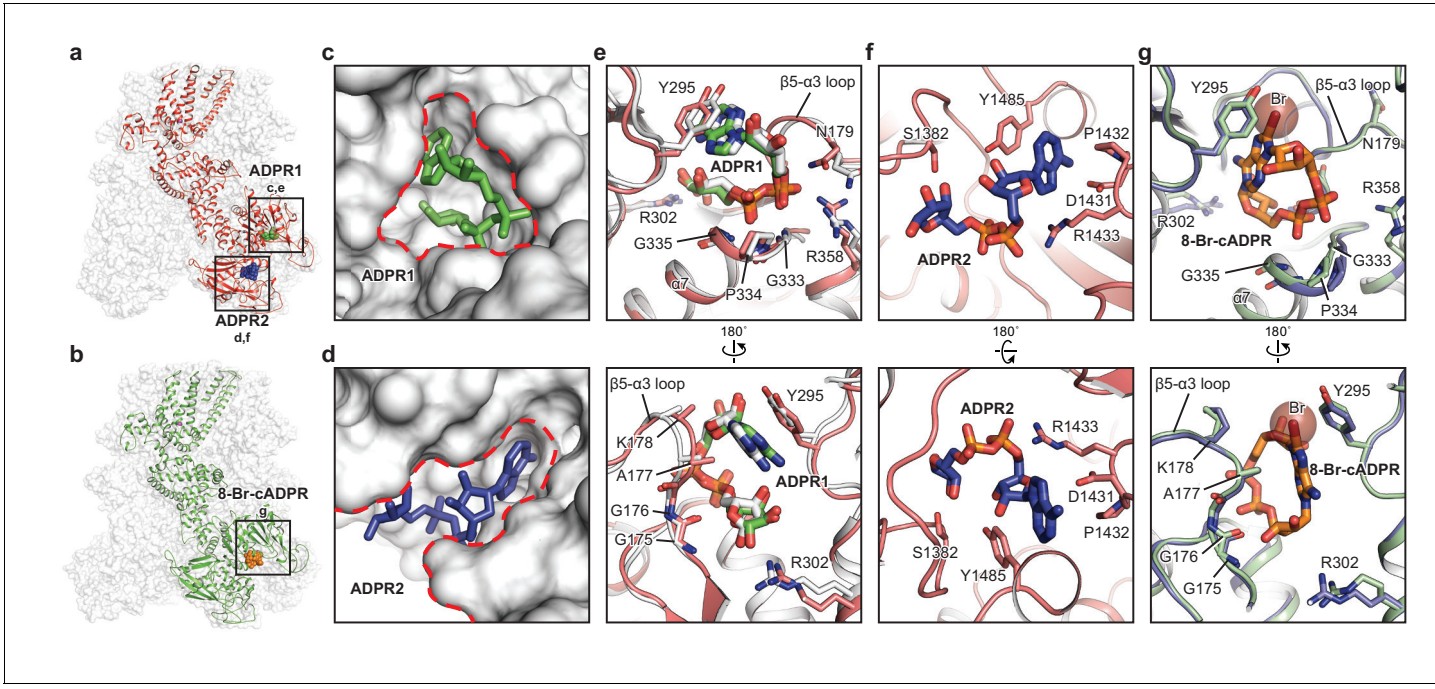

**Figure 2.** The ADPR1, ADPR2, and 8-Br-cADPR binding sites. The structures of the ADPR/Ca$^{2+}$-*hs*TRPM2 (**a**) and 8-Br-cADPR/Ca$^{2+}$-*hs*TRPM2 (**b**), with the locations of the ADPR1, ADPR2, and 8-Br-cADPR binding sites boxed. (**c–d**) Shapes of the ADPR1 (**c**) and ADPR2 (**d**) binding sites (outlined by red dashes). ADPR1 and ADPR2 are shown as sticks. (**e–g**) The ligand binding sites for ADPR1 (**e**), ADPR2 (**f**), and 8-Br-cADPR (**g**). The ligands, and key residues involved in ligand binding are shown as sticks. Superimposition of the human TRPM2 (protein in red and ADPR in green) with the zebrafish TRPM2 (white) shows that the ADPR1 sites are conserved in both organisms (**e**). Superimposition of the EDTA-*hs*TRPM2 with 8-Br-cADPR/Ca$^{2+}$-*hs*TRPM2 shows that binding of 8-Br-cADPR barely induces conformational change of the ADPR1 binding site. The bromine atom is shown as transparent sphere.

DOI: https://doi.org/10.7554/eLife.50175.008

The following figure supplement is available for figure 2:

**Figure supplement 1.** Secondary structure arrangement of *hs*TRPM2 and sequence alignment of TRPM2 from selected species.
DOI: https://doi.org/10.7554/eLife.50175.009

our 8-Br-cADPR/$Ca^{2+}$-*hs*TRPM2 structure, the 8-Br-cADPR molecule is observed only in the cleft of MHR1/2 domain (**Figure 2b**), but not in the NUDT9-H domain as previously proposed (**Eraslan et al., 2019**; **Kolisek et al., 2005**). This observation further supports the idea that the MHR1/2 domain is a key ligand-binding site not only for the agonist ADPR but also for the antagonist 8-Br-cADPR of cADPR.

## Ligand-binding sites

The presence of both ADPR and $Ca^{2+}$ initiates a rapid opening of *hs*TRPM2, which permits the flow of cations through the ion-conducting pore (**Csanády and Törocsik, 2009**; **McHugh et al., 2003**; **Perraud et al., 2001**; **Starkus et al., 2007**). Contrary to the consensus view that the NUDT9-H domain is the only binding site for ADPR and its derivatives, we found that the MHR1/2 domain binds both ADPR and 8-Br-cADPR, whereas NUDT9-H is only accessible to ADPR. Although both sites bind ADPR, the shapes of the two binding sites are distinct, which means that access is limited to ligands of a certain molecular geometry (**Figure 2a,c,d**).

The binding site in the bi-lobed MHR1/2 cleft is relatively deep, small, and circular. Therefore, ADPR1 is bent into a compact U shape to adapt to the MHR1/2 site (**Figure 2c**), similar to the finding in the *dr*TRPM2 structure (**Figure 2e**) (**Huang et al., 2018**). ADPR1 is clamped between Y295 and the loop connecting the β5 strand α3 helix (β5-α3 loop), with the terminal adenine and terminal ribose moieties close to each other (**Figure 2e**). Two phosphate groups interact with R358 and the N-terminus of the α7 helix, while the ribose moiety of the adenosine group coordinates with R302.

In contrast, the cleft in the NUDT9-H domain is wider, embraced by its cap and core regions on each side and the central β sheet on the bottom, allowing ADPR2 to nestle in an extended shape with the terminal adenine and terminal ribose moieties far apart (**Figure 2d**). The adenine moiety of ADPR2 stacks between Y1485 and D1431, while the α-phosphate group interacts with R1433 (**Figure 2f**). The β-phosphate group and the terminal ribose group are less well defined, presumably due to lack of interaction with the protein. ADPR2 in such an extended shape has been observed in other high-resolution structures such as the human protein ADP-ribosylargenine hydrolase (**Rack et al., 2018**).

The 8-Br-cADPR shares a similar shape with the U-shaped ADPR1 by connecting the terminal adenine and terminal ribose moieties, but it differs from the extended shape of ADPR2, thus fitting only into the MHR1/2 site (**Figure 2b**). While the residues interacting with 8-Br-cADPR, including Y295 and residues in the β5-α3 loop, are mostly the same as those interacting with ADPR1, a small displacement of the side chain of Y295 was observed due to the extra bromine atom of 8-Br-cADPR (**Figure 2g**).

To ground of our interpretations of the dual ADPR binding sites in the ligand-sensing domain of TRPM2 in the context of physiological function, we generated mutants of key residues involved in ADPR binding on both sites and we conducted inside-out patch-clamp experiments. Based on several lines of evidence, we concluded that MHR1/2 represents an orthosteric binding site in TRPM2 across all species. First, several mutants of the key residues in the binding site in the MHR1/2 domain either markedly decreased or abolished channel activation in response to ADPR/$Ca^{2+}$, which is caused by either altered affinity of ADPR to the MHR1/2 site, or altered allosteric transition from ligand binding to channel opening, or a combination of both (**Figure 3a–c**). These results indicate that the MHR1/2 binding site is crucial for channel gating of *hs*TRPM2, which agrees with previous functional studies on *hs*TRPM2 and *dr*TRPM2 (**Huang et al., 2018**; **Kashio et al., 2012**; **Luo et al., 2018**). Second, key residues responsible for ligand binding in the MHR1/2 site are highly conserved across species from invertebrates to mammals, whereas those in the NUDT9-H site are less conserved (**Figure 3d**, **Figure 2—figure supplement 1**). Third, ADPR-evoked channel activation of invertebrate *nv*TRPM2 is independent of its NUDT9-H domain, because truncation of that domain does not affect channel function (**Iordanov et al., 2019**; **Kühn et al., 2016**).

The next question is whether the NUDT9-H site acts as an allosteric binding site or as a second orthosteric binding site. Our electrophysiology data (**Figure 3a**), as well as previous reports by other labs (**Fliegert et al., 2017**; **Yu et al., 2017**), showed that mutations of the key residues in the NUDT9-H site markedly affected the channel function of *hs*TRPM2, indicating that this binding site is indispensable for *hs*TRPM2 activation. We suggest that the binding of ADPR to the NUDT9-H site assists channel opening in *hs*TRPM2, which will be discussed in the next section.

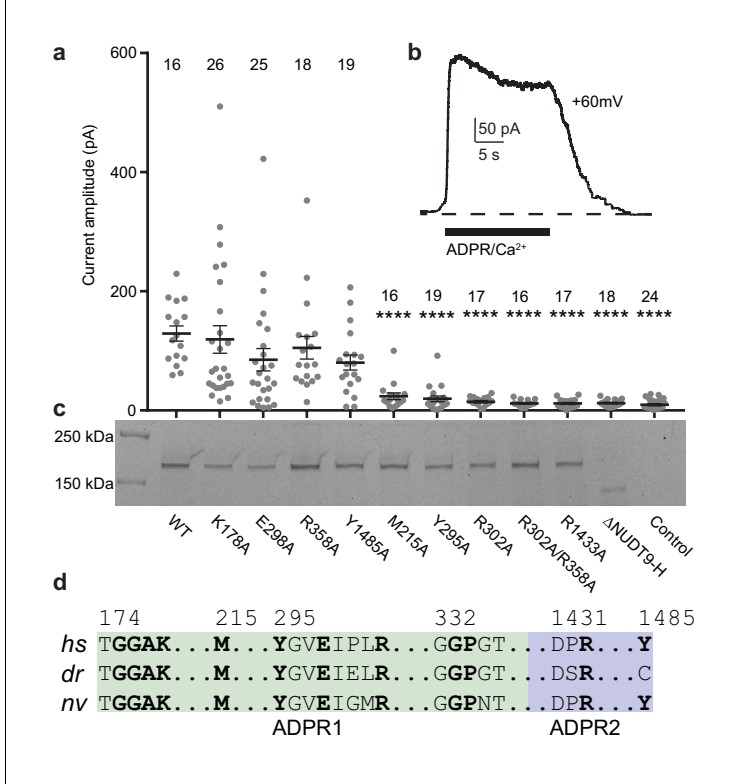

**Figure 3.** Residues and domain that are critical for ADPR-invoked channel activation. (**a**) Statistics of current amplitudes and cell numbers of inside-out patches pulled from HEK293 cells transfected with wild-type *hs*TPRM2, alanine mutants of key residues in the ADPR1 and ADPR2 sites, and a NUDT9-H-truncated construct, ΔNUDT9-H. Control experiments were carried out using non-transfected HEK293 cells. Mutants showing significantly smaller currents than the wild-type *hs*TRPM2 are labeled with asterisks (****: $p<0.0001$). Data represent the mean ± s. e. m. and numbers on the top of each column indicated the number of patches that recorded. Values from individual experiments are shown as filled circles. (**b**) Representative current in an inside-out patch pulled from HEK293 cell transfected with wild-type *hs*TRPM2 cDNA. (**c**) Surface expression profiles of *hs*TRPM2 constructs used in (**b**), detected by in-gel GFP fluorescence. Most of the mutants showed expression comparable to that of the wild-type *hs*TRPM2; the ΔNUDT9-H showed weaker but recognizable expression at lower molecular weight due to truncation of the NUDT9-H domain. (**d**) Sequence alignment of the ADPR1 and ADPR2 binding sites (*hs* = *Homo sapiens*; *dr* = *Danio rerio*; *nv* = *Nematostella vectensis*). Key residues for ligand binding are in bold.
DOI: https://doi.org/10.7554/eLife.50175.010

## Principles of agonist and antagonist actions in channel gating

The coexistence of three ligand-binding sites—a $Ca^{2+}$ binding site in the TMD and two ADPR binding sites in the ligand-sensing layer consisting of the MHR1/2 and NUDT9-H domains—endows *hs*TRPM2 a complicated gating mechanism. The ADPR- and ADPR/$Ca^{2+}$-*hs*TRPM2 structures differ only in the S1-S4 domain, suggesting that the conformational changes induced by $Ca^{2+}$ binding is restricted in the TMD and $Ca^{2+}$ binding facilitates the channel opening, which agrees with published structures of *dr*TRPM2 and *hs*TRPM4 (*Autzen et al., 2018*; *Huang et al., 2018*). Therefore, the intracellular domain is solely manipulated by the ligands binding in the MHR1/2 and NUDT9-H sites.

To understand their relationship with channel gating, we first compared the ligand-sensing layer of the three ligand-bound *hs*TRPM2 structures with the apo state structure. Despite a different relative positioning between the NUDT9-H and the MHR1/2 domains, the *hs*TRPM2 showed a rearrangement of the ligand-sensing layer upon binding of ADPR similar to that of *dr*TRPM2 (*Figure 4—figure supplement 1a–c*) (*Huang et al., 2018*). These conserved movements include a pendulum swing of the NUDT9-H domain toward pore axis, along with a contraction and clockwise rotation of the entire NUDT9-H layer, and an upward shift and an outward rotation of the MHR1/2 domain. By

contrast, binding of 8-Br-cADPR did not result in an obvious conformational change of these two domains (*Figure 4—figure supplement 1a,d*).

To explore the actions of agonists and antagonists, we compared the MHR1/2 domain and the NUDT9-H domain of *hs*TRPM2 in the absence or presence of ligands. ADPR binding into the cleft of MHR1/2 led to a clamshell closure similar to that observed in the *dr*TRPM2 structures (*Figure 4a*, *Figure 4—figure supplement 1e–f*). Binding of ADPR into the cleft of the NUDT9-H induced a rigid-body rotation of the core region toward the cap region, resulting in a bi-lobed domain closure (*Figure 4b*, *Figure 4—figure supplement 1h–i*). This is the first time that such a motion upon ligand binding has been observed for the NUDT9-H domain, which is likely conserved in its analog – the mitochondrial ADP-ribose pyrophosphatase, NUDT9 (*Shen et al., 2003*).

In contrast, the binding of 8-Br-cADPR barely produced any conformational changes (*Figure 4—figure supplement 1g,j*). Despite the fact that 8-Br-cADPR mimics U-shaped ADPR, the extra bromide atom is obstructed by Y295 (*Figure 2g*), preventing 8-Br-cADPR from going as deep as ADPR into the MHR1/2 cleft. As a result, the phosphate groups of 8-Br-cADPR block helix α7 from approaching, thus precluding clamshell closure (*Figure 4—figure supplement 1g*). This supports the concept that 8-Br-cADPR acts as an antagonist for cADPR in *hs*TRPM2, which inhibits the channel by occupying and keeping the MHR1/2 clamshell in an open conformation and stabilizing *hs*TRPM2 in an apo/resting-like state. Such action of an antagonist has been reported for other ion channels (*Mansoor et al., 2016*).

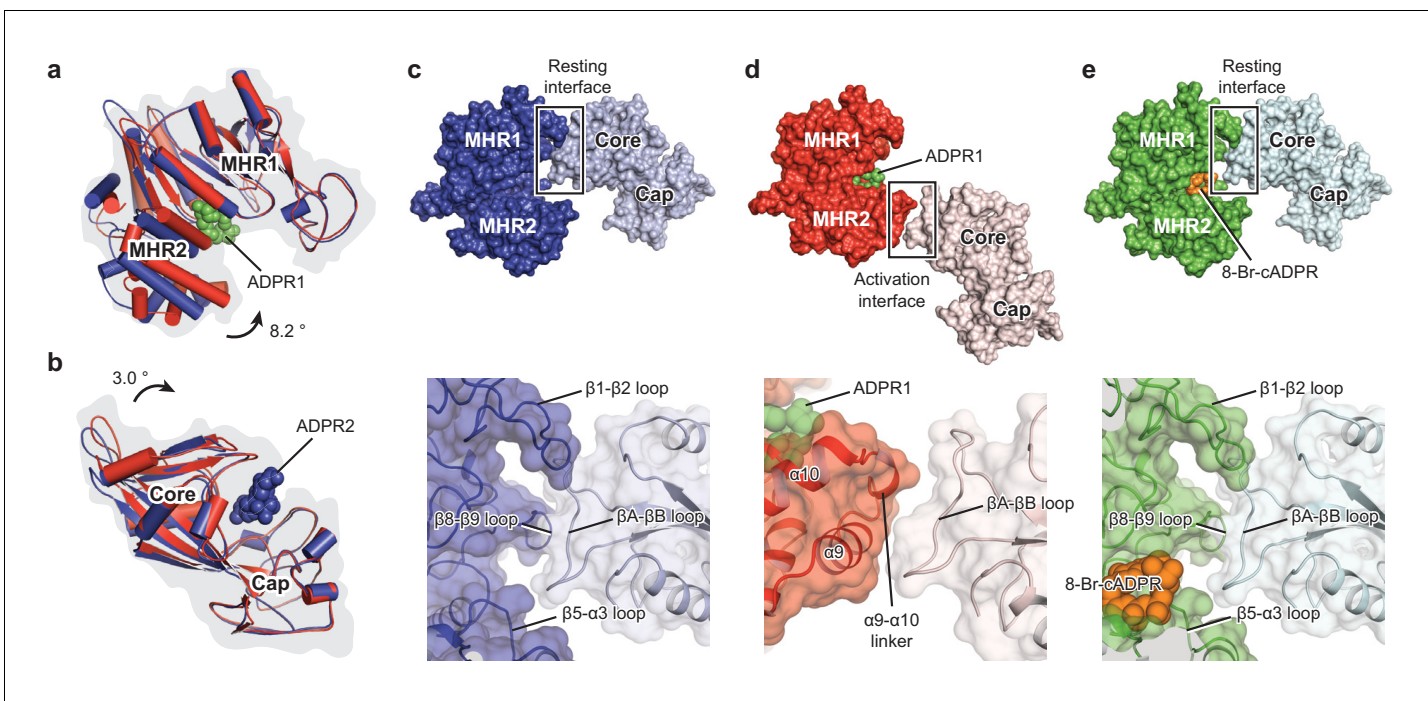

**Figure 4.** Domain rearrangement of the ligand-sensing layer – the MHR1/2 and NUDT9-H domains – upon binding of ADPR or 8-Br-cADPR. Comparison of (**a**) the MHR1/2 domains and (**b**) the NUDT9-H domains of EDTA-*hs*TRPM2 (blue) and ADPR/Ca²⁺-*hs*TRPM2 (red) by superimposition of the MHR1 domains or the cap regions. Domain closure was observed in the MHR1/2 and NUDT9-H domains upon ADPR binding. (**c**) In the apo resting state, the intersubunit interface (resting interface) is formed by the MHR1 domain and the adjacent core region of the NUDT9-H domain. The rectangle is enlarged in the lower image. (**d**) Upon binding of ADPR, the intersubmit interface (activation interface) is reorganized and is formed by the MHR2 domain and the adjacent core region. The two adjacent subunits are colored in dark or light colors. Key elements involved in intersubunit interfaces are labeled. (**e**) Binding of 8-Br-cADPR to the MHR1/2 domain does not result in domain rearrangement of the ligand-sensing layer relative to the apo state shown in panel c.

DOI: https://doi.org/10.7554/eLife.50175.011

The following figure supplement is available for figure 4:

**Figure supplement 1.** Conformational rearrangements in the ligand-sensing layer of *hs*TRPM2 upon binding of ligands.

DOI: https://doi.org/10.7554/eLife.50175.012

On the basis of structural comparison among the $hs$TPRM2 structures and the previously published functional studies on invertebrate TRPM2 (*Iordanov et al., 2019*; *Kühn et al., 2016*), we suggest that the MHR1/2 clamshell closure is the major driving force of channel opening by initiating motion in the intracellular domain which is transduced to the TMD, eventually causing channel opening in cooperation with $Ca^{2+}$. This raises a key question of why it is necessary to have a second ADPR molecule binding in NUDT9-H for channel activation.

To address this question, we inspected into the ligand-sensing domain upon binding of ADPR and found a striking interface rearrangement between NUDT9-H and the adjacent MHR1/2 domain (*Figure 4c–d*). In the absence of ADPR, their interface is made of extensive interactions between the core region of NUDT9-H and MHR1 (*Figure 4c*). Upon binding of ADPR, their interface in the apo resting state is disrupted, and a new interface is created in the activation state between the core region and MHR2 (*Figure 4d*). Accordingly, we call the interface between MHR1/2 and NUDT9-H a resting interface in the apo state and an activation interface in the ADPR-bound state. Because the core region of the NUDT9-H domain rotates toward the cap region upon binding of ADPR2, we suggest that the ADPR2 may play a role in disrupting the resting interface by pulling the core region away from the adjacent MHR1 domain, which in turn promotes the clamshell closure of MHR1/2 upon binding of ADPR1—a motion that requires flexibility of NUDT9-H that moves along with MHR2, thus assisting channel activation. The binding of 8-Br-cADPR (*Figure 4e*) did not produce any domain rearrangement relative to the apo state (*Figure 4c*).

## Channel activation and inhibition

The conformational change of the ligand-sensing layer upon binding of ADPR is transduced to the TMD through MHR3/4, whose motion is conserved between $hs$TRPM2 (*Figure 5—figure supplement 1*) and $dr$TRPM2 (*Huang et al., 2018*). To determine the functional states of the four $hs$TRPM2 structures, we looked at into their pore regions and compared their TMDs (*Figure 5a–g*). By comparing the distances between the $C\alpha$ of adjacent pore-restricting residues—N1049 and Q1053—with those in $dr$TRPM2 (*Huang et al., 2018*), all four showed an occluded pore, including the ADPR/ $Ca^{2+}$-$hs$TRPM2 structure (*Figure 5c*). The EDTA-$hs$TRPM2 and the 8-Br-cADPR/$Ca^{2+}$-$hs$TRPM2 structures (*Figure 5a,d*) were nearly identical, and they clearly represented an apo-resting state and an antagonist-bound inhibited closed state, respectively. In contrast, binding of ADPR caused large conformational changes throughout the protein relative to the apo state. Within the TMD specifically, although the pore domain showed little change, the S1-S4 domain underwent a remarkable clockwise rotation around the pore axis (*Figure 5e,f*). The subtle difference between the S1-S4 domains in ADPR- and ADPR/$Ca^{2+}$-$hs$TRPM2 was caused by the binding of $Ca^{2+}$ (*Figure 5g*). The $Ca^{2+}$ is coordinated by E843, Q846, N869, D872 and E1073, which are conserved among all the $Ca^{2+}$-sensitive TRPM channels (*Figure 5h*) (*Autzen et al., 2018*; *Huang et al., 2018*; *Yamaguchi et al., 2019*; *Yin et al., 2019a*). Because ADPR alone cannot active the TRPM2 channel in the absence of $Ca^{2+}$ (*McHugh et al., 2003*; *Sumoza-Toledo and Penner, 2011*), we designated the ADPR-$hs$TRPM2 structure as a ADPR-bound non-activated state.

The ADPR/$Ca^{2+}$-$hs$TRPM2 structure is closed, unlike the ADPR/$Ca^{2+}$-$dr$TRPM2 structure, which has a wide, open pore. A key element of the TRPM2 channel opening is the flipping of the S4-S5 linker from one side of the TRP helix to the other as a result of the synergetic actions of ADPR and $Ca^{2+}$, which is derived from our two $dr$TRPM2 structures (*Huang et al., 2018*). Indeed, the S4-S5 linkers in the ADPR/$Ca^{2+}$-$hs$TRPM2 structure (*Figure 5i,j*) are un-flipped, indicating that the TMD conformation represents either a pre-open or an inactive state. We favor the former, because the structure of ADPR/$Ca^{2+}$-$hs$TRPM2 is nearly identical with the non-activated ADPR-$hs$TRPM2. Moreover, the recently published two-fold symmetric $dr$TRPM2 adopts a hybrid of alternating non-flipped and flipped conformation of the S4-S5 linkers in TMD (*Yin et al., 2019b*), the former of which represents an intermediate state after ligand binding but prior to channel opening, which is similar to the TMD conformation of in our ADPR/$Ca^{2+}$-bound $hs$TRPM2 structure. This further supports that the ADPR/$Ca^{2+}$-bound $hs$TRPM2 structure represents a pre-open state.

We have noted that the published $hs$TRPM2 structure in the presence of $Ca^{2+}$ and ADPR at 6.4 Å overall resolution was reported as an open state (*Wang et al., 2018*). However, that interpretation is questionable, not only because most of the transmembrane domain was poorly defined in the cryo-EM map but also because the S4-S5 linker in the ADPR/$Ca^{2+}$-bound structure was clearly on the same side as in the apo resting structure (*Wang et al., 2018*).

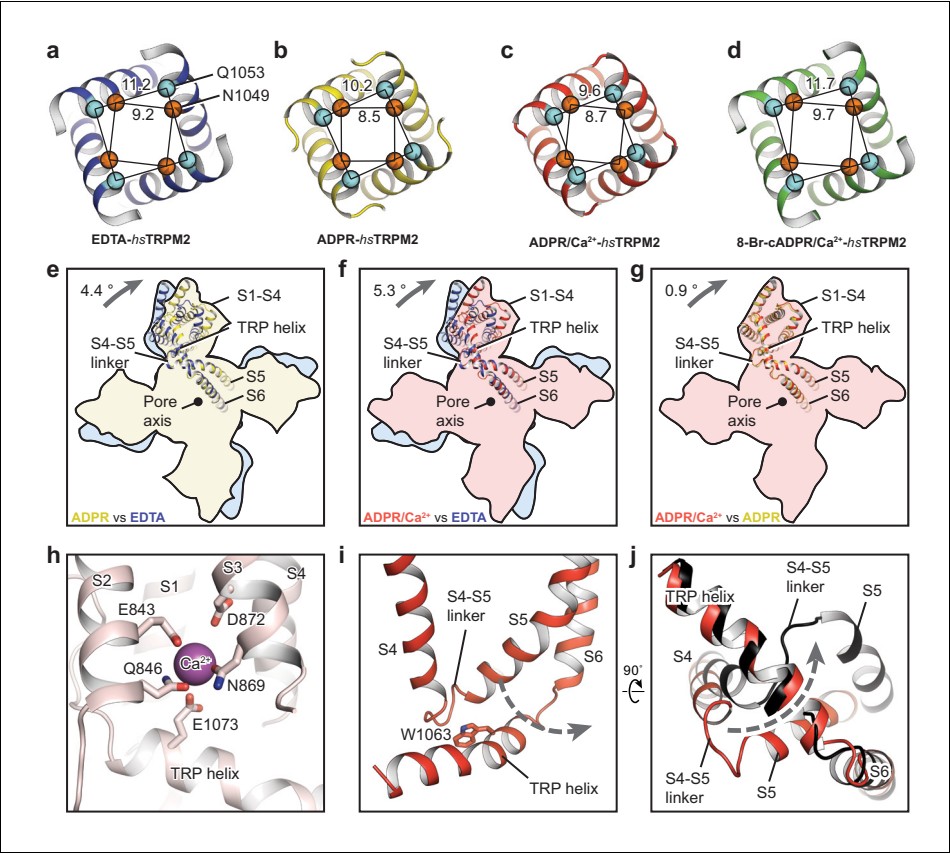

**Figure 5.** The ion-conducting pore. The gates of (**a**) EDTA-*hs*TRPM2, (**b**) ADPR-*hs*TRPM2, (**c**) ADPR/Ca²⁺-*hs*TRPM2, and (**d**) 8-Br-cADPR/Ca²⁺-*hs*TRPM2, viewed from the intracellular side. The distances between the Cα atoms of adjacent N1049 and the distances between the Cα atoms of adjacent Q1053 are indicated. (**e–g**) Comparison of the TMDs of EDTA-*hs*TRPM2 and ADPR-*hs*TRPM2 (**e**), EDTA-*hs*TRPM2 and ADPR/Ca²⁺-*hs*TRPM2 (**f**), and ADPR-*hs*TRPM2 and ADPR/Ca²⁺-*hs*TRPM2 (**g**) by superimposition of their pore domain of the tetramer. While the pore-lining S6 and S5 are well aligned, a clockwise rotation of the S1-S4 domain and TRP helix is observed from EDTA-*hs*TRPM2 and ADPR/Ca²⁺-*hs*TRPM2. (**h**) Calcium binding sites in the ADPR/Ca²⁺-*hs*TRPM2 structure. (**i**) The relative positioning between the S4-S5 linker and the TRP helix in ADPR/Ca²⁺-*hs*TRPM2, viewed parallel to the membrane. The dashed arrow illustrates the proposed movement of the S4-S5 linker from one side to the other of the TRP helix, which is required for channel opening upon binding of ADPR/Ca²⁺ that is observed in *dr*TRPM2 (*Huang et al., 2018*). (**j**) Comparison of the S4-S5 linkers in ADPR/Ca²⁺-*hs*TRPM2 (red) and ADPR/Ca²⁺-*dr*TRPM2 (black), viewed from the intracellular side. The two structures are superimposed using the TRP helix.
DOI: https://doi.org/10.7554/eLife.50175.013

The following figure supplement is available for figure 5:

**Figure supplement 1.** Conformational changes in the MHR3/4 domains upon binding of ADPR.
DOI: https://doi.org/10.7554/eLife.50175.014

## Discussion

A central concept of the relationship between ligand binding and channel gating in TRPM2 arises on the basis of our *hs*TRPM2 and *dr*TRPM2 structures, in which the functional state of the channel correlated with the conformation of the MHR1/2 domain; that is, the channel is in a ligand-free, apo (resting) state or an antagonist-bound inhibited state when MHR1/2 is open, and the channel is in an agonist-bound active state when the MHR1/2 is closed (*Figure 6*). The channel opening also requires the binding of Ca²⁺ in the TMD and a second ADPR molecule in the NUDT9-H domain. Unlike the consensus view of the NUDT9-H domain acting as the primary binding site for ADPR and its analogs, our data suggest that MHR1/2 represents an orthosteric binding site in TRPM2 across species, with key interaction residues conserved from invertebrates to mammals. TRPM2 is one example in which activation of the channel requires the synergetic binding of two ligands in three distinct sites.

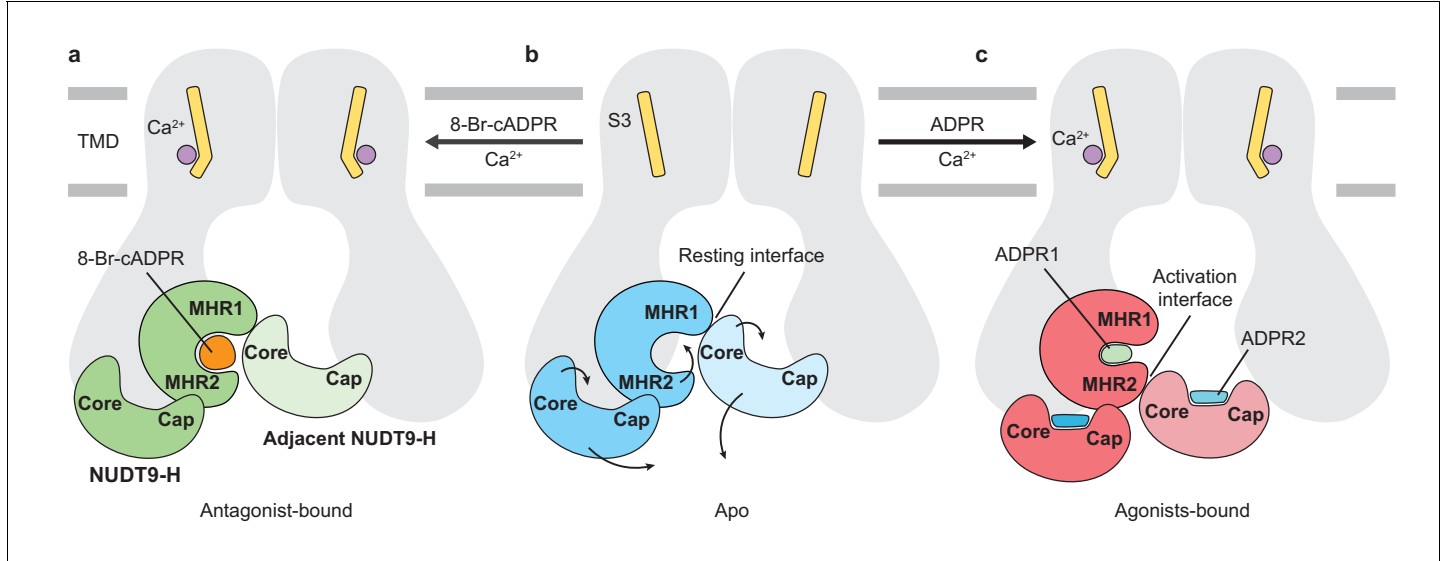

**Figure 6.** Schematic of ligand-sensing of *hs*TRPM2. (a–c) Conformational changes of *hs*TRPM2 among antagonist-bound (a), apo-resting (b), and agonist-bound (c) structures, focusing on the three ligand-binding sites. (a) 8-Br-cADPR binds only to the cleft of the MHR1/2 domain but not to the NUDT9-H domain. It inhibits the channel by stabilizing the MHR1/2 domain in an apo-like conformation. The $Ca^{2+}$ binding causes a tilting of the nearby S3 in the TMD, which is suggested to facilitate channel opening in the presence of agonist ADPR (*Csanády and Törocsik, 2009*; *McHugh et al., 2003*; *Starkus et al., 2007*). However, $Ca^{2+}$ alone is not sufficient to open the channel. (c) Conformational changes of *hs*TRPM2 upon binding of $Ca^{2+}$ and ADPR in triple binding sites with $Ca^{2+}$ bound nearby S3, the U-shaped ADPR1 (green) in the cleft of MHR1/2 domain, and the extended ADPR2 (blue) in the cleft of NUDT9-H domain. Binding of ADPR1 and ADPR2 induces bi-lobed domain closure of the MHR1/2 domain and the NUDT9-H domain, respectively. The MHR1/2 domain shows a counterclockwise rotation and the NUDT9-H domain swings toward the pore center. As a result, the resting interface is disrupted, and an activation interface is created between adjacent subunits. The movements induced by ADPR in the MHR1/2 domain and NUDT9-H domain are shown by arrows.

DOI: https://doi.org/10.7554/eLife.50175.015

The following figure supplement is available for figure 6:

**Figure supplement 1.** Comparison of the NUDT9-H domain with the NUDT9 enzyme.
DOI: https://doi.org/10.7554/eLife.50175.016

What then is the function of the characteristic NUDT9-H domain? It is indispensable for vertebrate TRPM2 channel activation, including human and zebrafish, but it is not required in invertebrates *nv*TRPM2 (*Kühn et al., 2016*). Comparison among the available TRPM2 structures reveals different interactions between NUDT9-H and MHR1/2 in human versus invertebrate TRPM2, which seemingly correlates with the functional importance of the NUDT9-H domain. In the *nv*TRPM2, the NUDT9-H domain likely lacks any interface with the rest of the protein, because it is completely invisible in the structure, despite the rest of the protein being well resolved (*Zhang et al., 2018*). By contrast, the *dr*TRPM2 has a visible but poorly defined NUDT9-H domain with a single major interface between the cognate MHR1/2 and NUDT9-H domains (*Huang et al., 2018*). Lastly, the *hs*TRPM2 has a well-defined NUDT9-H domain that has two major interfaces with the rest of protein, one with the cognate MHR1/2 and the other with the adjacent MHR1/2 domain. Such an incrementally tighter interaction between the NUDT9-H domain and the rest of protein is in harmony with the fact that the NUDT9-H domain gained more function along the evolution from invertebrates to mammals, such as temperature sensing, redox sensing, and channel gating, to endow TRPM2 with a polymodal nature. In parallel, the NUDT9-H domain has lost the ability to hydrolyze ADPR into AMP and ribose-5-phosphate, a function that exists only in invertebrate TRPM2 and NUDT9 (*Iordanov et al., 2016*; *Iordanov et al., 2019*; *Tóth et al., 2014*). A comparison between NUDT9-H in *hs*TRPM2 and NUDT9 (*Shen et al., 2003*) shows a wider opening of the cleft in NUDT9 (*Figure 6—figure supplement 1*), which is likely related to the fact that the core region in *hs*TRPM2 is restricted by the adjacent MHR1/2, whereas NUDT9 lacks such a restriction. Such a restriction for the NUDT9-H domain is

also missing in invertebrate TRPM2 such as *nv*TRPM2, perhaps giving rise to a similar cleft opening as in NUDT9 and thus preserved enzymatic activity (*Iordanov et al., 2019*; *Kühn et al., 2016*).

In summary, our human TRPM2 structures defined two distinctive ADPR binding sites, one in the MHR1/2 domain and one in the NUDT9-H domain. These structures demonstrate how incorporation of three different ligand binding sites, including the $Ca^{2+}$ site in the TMD, regulates channel function. Our work provides a clear structural explanation for ligand recognition, pharmacology, and the gating mechanism of the polymodal TRPM2 channel.

## Materials and methods

### Construct description, Expression and purification of full-length human TRPM2

The full-length human *TRPM2* gene (UniProtKB - O94759) was synthesized by Bio Basic and subcloned into a pEG BacMam vector with a twin-Strep tag, $His_8$ tag, enhanced green fluorescent protein (eGFP), and a thrombin cleavage site at the N terminus of the gene (*Goehring et al., 2014*; *Haley et al., 2019*). Plasmid DNA for the *hs*TRPM2 construct was transformed into DH5alpha competent cells, expanded into a large-scale bacterial culture, and isolated using EndoFree Plasmid kits (Qiagen). Purified plasmid DNA was mixed with PEI 25K (Polysciences) in a 3:1 ratio of PEI to DNA (w/w) and incubated at room temperature for 30 min. The PEI–DNA mixture was added to a suspension culture of HEK293 cells (ATCC Cat# CRL-11268, tested negative for mycoplasma contamination) at a density of 2.5–3.0 $\times$ $10^6$/mL for protein expression. After 12–24 hr post-transfection at 37°C, sodium butyrate (10 mM) was added to the suspension and the temperature was adjusted to 30°C to boost the protein expression. Seventy-two hours after transfection, HEK293 cells were harvested and washed with cold TBS buffer (150 mM NaCl, 20 mM Tris HCl, pH 8.0).

Harvested cells were lysed for 1 hr in 10 mM Tris-HCl pH 8.0 buffer containing a protease inhibitor cocktail (1 mM phenylmethylsulfonyl fluoride (PMSF), 2 mM pepstatin, 0.8 µM aprotinin, and 2 µg/ml leupeptin). After 1 hr of incubation, Tris-HCl pH 8.0 and NaCl were added to final concentrations of 20 mM and 150 mM, respectively. Lysed cells were incubated on ice for an additional 20 min. Cell debris and unlysed cells were removed by centrifugation at 2000 x *g* for 10 min. Ultracentrifugation was used to collect the membranes, using a 45 Ti rotor at 186,000 x *g* for 1 hr at 4°C (Beckman Coulter). The collected membranes were homogenized using a Dounce homogenizer in TBS buffer containing a protease inhibitor cocktail and 2 mM 2-mercaptoethanol. Solubilization of the membrane was performed by using 10 mM glyco-diosgenin (Anatrace) for 1 hr at 4°C before ultracentrifugation for 30 min at 186,000 x *g*. The supernatant was transferred out of centrifuge tubes and incubated with Talon resin (Clontech) for 2 hr. Talon resin was washed with six bed volumes of TBS buffer containing 0.2 mM glyco-diosgenin, 10 mM imidazole, and 2 mM 2-mercaptoethanol. The protein was eluted with TBS buffer containing 0.2 mM glyco-diosgenin, 250 mM imidazole, and 2 mM 2-mercaptoethanol. The protein eluate was collected, concentrated, and loaded onto a Superose 6 column (GE Healthcare) using TBS buffer containing 0.2 mM glyco-diosgenin and 5 mM 2-mercaptoethanol. The peak fractions were combined and concentrated to 6.0–7.0 mg/mL using a 100 kDa concentrator (Millipore).

### Electron microscopy sample preparation and data acquisition

Quantifoil holey carbon grids (Au 1.2/1.3 µm size/hole space, 300 mesh) were glow-discharged for 30 s before preparation. Purified and concentrated *hs*TRPM2 protein was incubated with either 1 mM EDTA; 1 mM ADPR (Sigma-Aldrich); 1 mM ADPR (Sigma-Aldrich) plus 1 mM $CaCl_2$; or 1 mM 8-Br-cADPR (Santa Cruz Biotechnology) plus 1 mM $CaCl_2$. After incubation, 2.5 µl of protein sample was added to the carbon face of the grids and blotted for 2 s with a 5 s waiting time. The grids were plunge-frozen into liquid ethane and then cooled by liquid nitrogen using a Vitrobot Mark III held at 18°C and 100% humidity.

Images were obtained using an FEI Titan Krios electron microscope operating at 300 kV with a nominal magnification of 130,000. Images were recorded by a Gatan K2 Summit direct electron detector, which operated in super-resolution counting mode and with a binned pixel size of 1.074 Å. Each image was dose-fractionated to 40 frames, with a total exposure time of 8 s and 0.2 s per frame. The dose rate was 6.76 $e^-Å^{-2}$ $s^{-1}$ for each image. Images were recorded using an automated

acquisition program (SerialEM) (*Mastronarde, 2005*). Nominal defocus values varied between −1.3 to −1.9 μm.

## Electron microscopy data processing

Using MotionCor2 (*Zheng et al., 2017*), images were motion-corrected, summed, and 2 × 2 binned in Fourier space. Defocus values were estimated using Gctf (*Zhang, 2016*). Particles were then picked using Gautomatch (http://www.mrc-lmb.cam.ac.uk/kzhang/Gautomatch/) and subjected to an initial reference-free 2D classification using RELION (*Scheres, 2012*). Nine representative 2D class averages were selected to use as templates for automated particle-picking for the entire data set using Gautomatch. The auto-picked particles were then visually checked in order to remove false positives. Several rounds of 2D classification were performed to further clean up the selected particles using RELION (*Scheres, 2012*). The initial reconstruction was obtained using cryoSPARC (*Punjani et al., 2017*). The particles were then placed into ten classes using the 3D classification function in RELION (*Scheres, 2012*), with the initial reconstruction low-pass-filtered to 50 Å as a reference model. Particles from classes showing high-resolution features were combined and refined with C4 symmetry using RELION (*Scheres, 2012*) (*Figure 1—figure supplement 1*). The final resolutions reported in *Table 1* are based on the gold standard Fourier shell correlation 0.143 criteria. A soft mask (6.4 Å extended from the reconstruction with an additional 6.4 Å cosine soft edge, low-pass-filtered to 15 Å) was applied to the two half maps in order to calculate the Fourier shell correlation plot. Local resolutions were estimated using Bsoft (*Heymann, 2018*).

**Table 1.** Statistics of 3D reconstruction and model refinement.

| Data collection/processing | EDTA-*hs*TRPM2 | ADPR-*hs*TRPM2 | ADPR/Ca$^{2+}$-*hs*TRPM2 | 8-Br-cADPR/Ca$^{2+}$-*hs*TRPM2 |
|---|---|---|---|---|
| Microscope | Titan Krios (FEI) | Titan Krios (FEI) | Titan Krios (FEI) | Titan Krios (FEI) |
| Voltage (kV) | 300 | 300 | 300 | 300 |
| Defocus range (μM) | 1.0–2.5 | 1.0–2.5 | 1.0–2.5 | 1.0–2.5 |
| Exposure time (s) | 8 | 8 | 8 | 8 |
| Dose rate (e$^-$/Å$^2$/s) | 6.8 | 6.8 | 6.8 | 6.8 |
| Number of frames | 40 | 40 | 40 | 40 |
| Pixel size (Å) | 1.076 | 1.076 | 1.076 | 1.076 |
| Particles picked | 783,885 | 537,671 | 2,342,060 | 1,822,211 |
| Particles 2D | 415,415 | 346,032 | 759,700 | 729,269 |
| Particles refined | 161,360 | 117,350 | 287,184 | 102,259 |
| Resolution (Å) | 3.3 | 4.4 | 3.7 | 3.7 |
| FSC threshold | 0.143 | 0.143 | 0.143 | 0.143 |
| Resolution range (Å) | 322.2–3.3 | 322.2–4.4 | 322.2–3.7 | 322.2–3.7 |
| **Model statistics** | | | | |
| Number of atoms | 37816 | 37308 | 37408 | 38492 |
| Protein | 37816 | 37020 | 37116 | 38344 |
| Ligand | 0 | 288 | 292 | 148 |
| **r.m.s. deviations** | | | | |
| Bond length (Å) | 0.007 | 0.008 | 0.009 | 0.006 |
| Bond angle (°) | 0.907 | 1.015 | 1.051 | 0.941 |
| **Ramachandran plot** | | | | |
| Favored (%) | 92.18 | 91.59 | 91.53 | 94.33 |
| Allowed (%) | 7.59 | 8.02 | 8.08 | 5.44 |
| Disallowed (%) | 0 | 0 | 0 | 0 |
| Rotamer outlier (%) | 0.35 | 1.40 | 1.38 | 1.13 |

DOI: https://doi.org/10.7554/eLife.50175.017

## Model building and structural determination

Models for human TRPM2 were built in Coot using the zebrafish TRPM2 structure as a reference (RCSB Protein Data Bank (PDB) ID: 6DRK and 6DRJ) (*Huang et al., 2018*). The initial models were then subjected to real space refinement using phenix.real_space_refine (*Afonine et al., 2012*) with secondary-structure restraints. The refined model was further manually examined and adjusted in Coot (*Emsley et al., 2010*). For validation of the refined structure, Fourier shell correlation curves were applied to calculate the difference between the final model and electron microscopy map by PHENIX comprehensive validation (cryo-EM) (*Afonine et al., 2018*). The geometries of the atomic models were evaluated using MolProbity (*Williams et al., 2018*) in PHENIX suite (*Adams et al., 2010*). All figures were prepared using UCSF Chimera (*Pettersen et al., 2004*) and ChimeraX (*Goddard et al., 2018*) and PyMOL (https://pymol.org).

## Electrophysiology

HEK293 cells were transfected using Lipofectamine 2000 (Thermo Fisher) according to the manufacturer's protocol. Transfected cells were plated in a 24-well plate and incubated at 37˚C. The surface expression of WT human TRPM2 and mutants were evaluated by Pierce cell surface protein isolation kit (Thermo Fisher) according to the manufacturer's protocol, except that the protein was eluted by TBS buffer supplemented with 50 mM Dithiothreitol (DTT) and detected by in-gel fluorescence. Currents from inside-out patches were recorded 12–24 hr post transfection and the recordings were performed by using a HEKA EPC-10 amplifier set to room temperature with a holding potential of +60 mV. The patch pipettes used for these experiments were filled with internal solution that consisted of 150 mM NaCl, 3 mM KCl, and 10 mM HEPES (pH 7.4, adjusted with NaOH). The bath solution was the same as the internal solution. A bath solution with 0.1 mM ADPR and 1 mM $CaCl_2$ was used to activate the *hs*TPRM2 channel expressed in HEK293 cells. To change the solution, a two-barrel theta-glass pipette controlled manually was used. Data was acquired at 10 kHz using Patchmaster software (HEKA) and for display purposes, data was digitally filtered at 100 Hz and down sampled by a factor of 10. Statistical analysis was done by GraphPad Prism (GraphPad Software), data was reported as mean ± s. e. m. and analyzed using an unpaired *t* - test.

## Data availability

The cryo-EM density map and coordinates of the *hs* TRPM2 apo state, the ADPR-bound state, the ADPR/$Ca^{2+}$-bound state, and the 8-Br-cADPR/$Ca^{2+}$-bound state have been deposited in the Electron Microscopy Data Bank (EMDB) under accession numbers EMD-20478, EMD-20479, EMD-20480 and EMD-20482 and in the Research Collaboratory for Structural Bioinformatics Protein Data Bank under accession codes 6PUO, 6PUR, 6PUS and 6PUU.

## Acknowledgements

We thank G Zhao and X Meng for the support with data collection at the David Van Andel Advanced Cryo-Electron Microscopy Suite. We appreciate the HPC team in VARI for computational support. We thank W Sun (Janelia Research Campus) for his helpful discussion on electrophysiology experiments. We thank D Nadziejka for technical editing. JD is supported by a McKnight Scholar Award, a Klingenstein-Simon Scholar Award and the National Institutes of Health (NIH) (grant 1R01NS111031-01; JD).

## Additional information

### Funding

| Funder | Grant reference number | Author |
| --- | --- | --- |
| Esther A. and Joseph Klingenstein Fund | 2019 class | Juan Du |
| McKnight Endowment Fund for Neuroscience | 2019 class | Juan Du |
| National Institutes of Health | R01NS111031 | Juan Du |

The funders had no role in study design, data collection and interpretation, or the decision to submit the work for publication.

## Author contributions

Yihe Huang, Data curation, Software, Formal analysis, Validation, Visualization, Methodology, Writing—review and editing; Becca Roth, Data curation, Validation, Methodology, Writing—review and editing; Wei Lü, Conceptualization, Resources, Data curation, Software, Formal analysis, Supervision, Validation, Investigation, Visualization, Methodology, Writing—original draft, Project administration, Writing—review and editing; Juan Du, Conceptualization, Resources, Data curation, Software, Formal analysis, Supervision, Funding acquisition, Validation, Investigation, Visualization, Methodology, Writing—original draft, Project administration, Writing—review and editing

## Author ORCIDs

Wei Lü https://orcid.org/0000-0002-3009-1025
Juan Du https://orcid.org/0000-0003-1467-1203

## Decision letter and Author response

Decision letter https://doi.org/10.7554/eLife.50175.028
Author response https://doi.org/10.7554/eLife.50175.029

# Additional files

## Supplementary files

• Transparent reporting form
DOI: https://doi.org/10.7554/eLife.50175.018

## Data availability

All the cryo-EM data generated in this study have been deposited to PDB and EMDB databank.

The following datasets were generated:

| Author(s) | Year | Dataset title | Dataset URL | Database and Identifier |
|---|---|---|---|---|
| Du J, Lu W, Huang Y | 2019 | Human TRPM2 in the apo state | http://www.rcsb.org/structure/6PUO | Protein Data Bank, 6PUO |
| Du J, Lu W, Huang Y | 2019 | Human TRPM2 bound to ADPR and calcium | http://www.rcsb.org/structure/6PUS | Protein Data Bank, 6PUS |
| Du J, Lu W, Huang Y | 2019 | Human TRPM2 bound to ADPR | http://www.rcsb.org/structure/6PUR | Protein Data Bank, 6PUR |
| Du J, Lu W, Huang Y | 2019 | Human TRPM2 bound to 8-Br-cADPR and calcium | http://www.rcsb.org/structure/6PUU | Protein Data Bank, 6PUU |

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
