## [Decision Letter]

Thank you for submitting your article "Ligand recognition and gating mechanism through three ligand-binding sites of human TRPM2 channel" for consideration by *eLife*. Your article has been reviewed by three peer reviewers, including Leon D Islas as the Reviewing Editor and Reviewer #1, and the evaluation has been overseen by Olga Boudker as the Senior Editor.

The reviewers have discussed the reviews with one another and the Reviewing Editor has drafted this decision to help you prepare a revised submission.

Summary:

The reviewers find that the present manuscript contains great quality data and that it is an important contribution. However they also find that in its present form, some of the conclusions are not supported by the presentation and that some results are over interpreted an overhyped. The authors use phrases like: "unprecedented and extremely complicated", which should be avoided. A more balanced presentation is required as well as attention to the main points below.

Essential revisions:

1) The reviewers find that the main conclusions about the roles of the two ADPR sites are not well substantiated. In particular no experiments regarding the open probabilities reached in the presence of one or two binding sites is presented. Also, the main conclusion of the paper that the ADPR1 site (MHR1/2) is the main site and the ADPR2 site (NUDH9H) is the secondary site stems from the fact that the inhibitor 8-Br-cADPR binds to the ADPR1 site, not the ADPR2 site. The authors indicate that 8-Br-cADPR acts as a competitive inhibitor. However, the authors failed to note that the Penner group showed that 8-Br-cADPR only inhibits cADPR, not ADPR (Kolisek, 2005). On the contrary, the Penner group found that co-application of ADPR and 8-Br-cADPR appeared to enhance the response of ADPR to some degree. Furthermore, cADPR is a very weak partial agonist, as its potency and efficacy are significantly lower than those of ADPR (> 50-fold). Therefore, the authors need to perform the similar experiment of co-application of 8-Br-cADPR and ADPR to see if there is any dose-dependent inhibition of 8-Br-cADPR. The authors need to do a better job of siting previous literature, especially functional studies, within context.

2) At the end of the first paragraph of the subsection “Channel activation and inhibition”, assignment of the state of the ADPR-bound structure as partially-activated state is inappropriate, as the pore is closed. "Partially activated" implies that it is partially open so that ion can permeate through the channel. "Pre-open state" is more appropriate for the ADPR-bound structure.

3) In the second paragraph of the subsection “Channel activation and inhibition”, assignment of the ADPR/Ca^2+^ bound structure as a pre-open state might be fine, but the authors' reasoning that "because ADPR-bound state is partially activated the ADPR/Ca^2+^ structure is pre-open" is not appropriate.

4) The authors reveal a second ADPR binding site in the NUDT9-H domain of hsTRPM2. In contrast, ADPR binding was not observed in the NUDT9-H domain of the ADPR/Ca^2+^-drTRPM2 structure in their previous study. Does the local conformation of the NUDT9-H in drTRPM2 differ from the corresponding ADPR2 site in the ADPR/Ca^2+^-hsTRPM2 structure? How would structural comparisons suggest the differential binding of ADPR in the NUDT9-H domain between the two species? In general, an in depth comparison of the present structures with previous structures from other organisms is need in order to better present the different conformations seen in the *H. sapiens* structures.

5) The current cryo-EM density figures (Figure 1—figure supplement 3E-H) for the ligands are too small. Also, half-maps for all the ligands are not provided, including ADPR1, ADPR2, and 8-Br-cADPR. Cryo-EM density for these ligands as well as their half-maps should be shown in a separate supplementary figure.

6) For the model-map FSC figures (Figure 1—figure supplement 2), FSC curves calculated between each half-map and the model should be provided as well.

7) The authors perform 3d reconstructions with C4 symmetry imposed. Have authors performed asymmetric (C1) reconstruction to ensure that all the reconstructions assume strict C4 symmetry?

8) Regarding the electrophysiological experiments with the presumed binding site mutants, reduction of the ionic current magnitude is hard to interpret. Is the affinity being affected or is the allosteric transition (s) leading to opening? Is the reduction of current observed in some mutants due to a shift of the dose response or are the channels completely irresponsive to ADPR? A dose response or at least the response to more than one concentration of ADPR should be obtained in order to provide a less ambiguous interpretation of these data. If these data cannot be obtained in reasonable time, at least the conclusion that these residues directly affect binding should be toned down and other different possibilities for the effects should be discussed.

9) Authors should provide their cryo-EM maps as well as coordinates.

---

## [Author Response]

Essential revisions:1) The reviewers find that the main conclusions about the roles of the two ADPR sites are not well substantiated. In particular no experiments regarding the open probabilities reached in the presence of one or two binding sites is presented. Also, the main conclusion of the paper that the ADPR1 site (MHR1/2) is the main site and the ADPR2 site (NUDH9H) is the secondary site stems from the fact that the inhibitor 8-Br-cADPR binds to the ADPR1 site, not the ADPR2 site. The authors indicate that 8-Br-cADPR acts as a competitive inhibitor. However, the authors failed to note that the Penner group showed that 8-Br-cADPR only inhibits cADPR, not ADPR (Kolisek, 2005). On the contrary, the Penner group found that co-application of ADPR and 8-Br-cADPR appeared to enhance the response of ADPR to some degree. Furthermore, cADPR is a very weak partial agonist, as its potency and efficacy are significantly lower than those of ADPR (> 50-fold). Therefore, the authors need to perform the similar experiment of co-application of 8-Br-cADPR and ADPR to see if there is any dose-dependent inhibition of 8-Br-cADPR. The authors need to do a better job of siting previous literature, especially functional studies, within context.

We thank the reviewers for this comment. We have now clarified in the Introduction and throughout the text that 8-Br-cADPR is an antagonist only for cADPR but not for ADPR and have cited the corresponding work from Penner’s group. We agree with reviewer that it is not conclusive to define the MHR1/2 as the primary site and the NUDT9-H as the secondary site. We have thus removed it from the text. Instead, we conclude that MHR1/2 represents an orthosteric binding site in TRPM2 across all species, which is explained in the fifth paragraph of the subsection “Ligand-binding sites”. It is worth noting that the Penner group found co-application of ADPR and 8-Br-cADPR did not always enhance the response of ADPR (Kolisek, 2005), on which there is currently no mechanistical understanding. In addition, we did not observe inhibition effect of 8-Br-cADPR for ADPR-evoked current either, in line with the findings by Penner’s group.

The open probability of wild type *hs*TRPM2 has been reported to be 0.67 in the presence of 100 µM ADPR and 1 µM free CalCl_2_ in an inside-out configuration (Fliegert, Nat Chem Biol. 2017). Based on our electrophysiology experiments, both binding sites are indispensable for channel activation, and no current can be recorded if one of the two ADPR site is “knocked out” (Figure 3, 100 µM ADPR and 1 mM Ca^2+^, inside-out patch). We agree with the reviewer that the apparently abolished channel activation at the electrophysiology experiment condition in this manuscript may be caused by altered ligand affinity, or altered transition from ligand binding to channel opening, or a combination of both. We have thus revised the text accordingly (see the aforementioned paragraph). However, before a Po can be measured, it requires systematical examination whether the channel can still be activated in the presence of one ADPR site, which is out of the scope of the current study and is not required to support our suggestion that MHR1/2 represents an orthosteric binding site in TRPM2 across all species in the revised manuscript.

2) At the end of the first paragraph of the subsection “Channel activation and inhibition”, assignment of the state of the ADPR-bound structure as partially-activated state is inappropriate, as the pore is closed. "Partially activated" implies that it is partially open so that ion can permeate through the channel. "Pre-open state" is more appropriate for the ADPR-bound structure.3) In the second paragraph of the subsection “Channel activation and inhibition”, assignment of the ADPR/Ca^2+^ bound structure as a pre-open state might be fine, but the authors' reasoning that "because ADPR-bound state is partially activated the ADPR/Ca^2+^ structure is pre-open" is not appropriate.

We agree with the reviewers that it is not appropriate to assign the ADPR-bound structure as partially-activated state. Because both ADPR and Ca^2+^ are absolutely required to activate the TRPM2, we believe it is appropriate to call it “ADPR-bound non-activated state”. We have also revised the reasoning according to the reviewers’ suggestion: “because the overall structure of ADPR/Ca^2+^-*hs*TRPM2 is nearly identical with the structure of non-activated ADPR-*hs*TRPM2.”

4) The authors reveal a second ADPR binding site in the NUDT9-H domain of hsTRPM2. In contrast, ADPR binding was not observed in the NUDT9-H domain of the ADPR/Ca^2+^-drTRPM2 structure in their previous study. Does the local conformation of the NUDT9-H in drTRPM2 differ from the corresponding ADPR2 site in the ADPR/Ca^2+^-hsTRPM2 structure? How would structural comparisons suggest the differential binding of ADPR in the NUDT9-H domain between the two species? In general, an in depth comparison of the present structures with previous structures from other organisms is need in order to better present the different conformations seen in the *H. sapiens* structures.

We fully agree with the reviewers that it would be insightful to compare structures of the same protein from different organisms. TRPM2 is a large and complex protein, and there are inherent local structural differences, e.g. secondary structures, between *hs*TRPM2 and *dr*TRPM2 which share a sequence similarity of less than 65%. Therefore, direct superimposing domain structures of the two species might not be sufficient to show the conformational differences which could be easily overwhelmed by the inherent structural differences between the two species. Nevertheless, the trends of motion induced by ligand binding are conserved for both proteins. Therefore, we believe it would be more meaningful to show these conformational changes along with measurements separately for each species. Several figures in this manuscript (Figure 2E, Figure 4A, Figure 5H-J, and Figure 1—figure supplement 5, Figure 4—figure supplement 1, Figure 5—figure supplement 1, Figure 6—figure supplement 1) are prepared in almost identical views to the ones in our *dr*TRPM2 paper (Huang, 2018), which nicely show the conserved motion upon ligand binding.

Unlike in *hs*TRPM2, the NUDT9-H domain in the *dr*TRPM2 structures are very flexible due to a lack of extensive interface with the rest of protein. Thus, we were only able to rigid-body fit a NUDT9-H homology model, which is generated using the crystal structure of human NUDT9, into the cryo-EM maps of EDTA-*dr*TRPM2 and ADPR/Ca^2+^-*dr*TRPM2. We were unable to reliably tell if there is difference between the NUDT9-H domain in EDTA-*dr*TRPM2 and ADPR/Ca^2+^-*dr*TRPM2, which has been clearly stated in our previous paper (Huang, 2018). Due to the same reason, we think it might be misleading to directly compare the structures of NUDT9-H of *hs*TRPM2 and *dr*TRPM2. Nevertheless, the rearrangement of the ligand-sensing layer upon binding of ADPR is conserved between two species, which is shown in Figure 4—figure supplement 1 of this manuscript and Figure 3 in Huang, 2018.

5) The current cryo-EM density figures (Figure 1—figure supplement 3E-H) for the ligands are too small. Also, half-maps for all the ligands are not provided, including ADPR1, ADPR2, and 8-Br-cADPR. Cryo-EM density for these ligands as well as their half-maps should be shown in a separate supplementary figure.

We have now included a new supplementary figure showing enlarged densities for ADPR1, ADPR2, 8-Br-cAPPR along with their half-maps (Figure 1—figure supplement 4).

6) For the model-map FSC figures (Figure 1—figure supplement 2), FSC curves calculated between each half-map and the model should be provided as well.

We have included model-map FSC curves between each half-map and model (Figure 1—figure supplement 2), which indicate no overfitting of the atomic models.

7) The authors perform 3d reconstructions with C4 symmetry imposed. Have authors performed asymmetric (C1) reconstruction to ensure that all the reconstructions assume strict C4 symmetry?

We have indeed performed C1 refinement prior to C4 refinement. The C1 maps are very similar to the C4 maps despite lower resolutions. We did not observe C2-symmeric features in the C1 maps as observed in the C2-symmetric *dr*TRPM2 (Yin et al., 2019).

8) Regarding the electrophysiological experiments with the presumed binding site mutants, reduction of the ionic current magnitude is hard to interpret. Is the affinity being affected or is the allosteric transition (s) leading to opening? Is the reduction of current observed in some mutants due to a shift of the dose response or are the channels completely irresponsive to ADPR? A dose response or at least the response to more than one concentration of ADPR should be obtained in order to provide a less ambiguous interpretation of these data. If these data cannot be obtained in reasonable time, at least the conclusion that these residues directly affect binding should be toned down and other different possibilities for the effects should be discussed.

We thank the reviewers for this comment that a reduction of current amplitude may be caused by different reasons. We have thus revised the text accordingly and added a discussion regarding possible reasons for the reduction of current.

“First, several mutants of the binding site in the MHR1/2 domain either markedly decreased or abolished channel activation in response to ADPR/Ca^2+^, which is caused by either altered affinity of ADPR to the MHR1/2 site, or altered allosteric transition from ligand binding to channel opening, or a combination of both (Figures 3A-C)”.

9) Authors should provide their cryo-EM maps as well as coordinates.

We have now provided the EMDB and PDB codes for the maps and coordinates in the section of Data Availability.